# Comparative Analysis and Phylogenetic Relationships of *Ceriops* Species (Rhizophoraceae) and *Avicennia lanata* (Acanthaceae): Insight into the Chloroplast Genome Evolution between Middle and Seaward Zones of Mangrove Forests

**DOI:** 10.3390/biology11030383

**Published:** 2022-02-28

**Authors:** Panthita Ruang-areerate, Thippawan Yoocha, Wasitthee Kongkachana, Phakamas Phetchawang, Chatree Maknual, Wijarn Meepol, Darunee Jiumjamrassil, Wirulda Pootakham, Sithichoke Tangphatsornruang

**Affiliations:** 1National Omics Center, National Science and Technology Development Agency (NSTDA), Pathum Thani 12120, Thailand; panthita.rua@nstda.or.th (P.R.-a.); thippawan.yoo@nstda.or.th (T.Y.); wasitthee.kon@nstda.or.th (W.K.); phakamas.phe@ncr.nstda.or.th (P.P.); 2Department of Marine and Coastal Resources, 120 The Government Complex, Chaengwatthana Rd., Thung Song Hong, Bangkok 10210, Thailand; c_maknual@hotmail.com; 3Department of Marine and Coastal Resources, Ranong Mangrove Forest Research Center, Tambon Ngao, Muang District, Ranong 85000, Thailand; wijarn.meepol@yahoo.com; 4Marine and Coastal Resources Office 5, 199/6 Khanom, Khanom, Nakhon Si Thammarat 80210, Thailand; darunee_ji@hotmail.com

**Keywords:** *Ceriops*, *Avicennia*, mangrove, chloroplast genome, plastid, comparative analysis, phylogenetic relationships

## Abstract

**Simple Summary:**

We sequenced the complete chloroplast genomes of three *Ceriops* species (*C. decandra*, *C. zippeliana*, and *C. tagal*) and *Avicennia lanata* and performed comparative analyses among them. All chloroplast genomes have a circular quadripartite structure containing LSC, SSC, and two IR regions. The *rpl32* gene was lost in *C. zippeliana*, and the *infA* gene was present in only *A. lanata*. Comparative genome analysis showed that the IR contraction or expansion events resulted in the differentiation of three genes and pseudogenes. Additionally, repeats and SSRs were identified and compared among them and other relative mangrove species. The phylogenetic analysis strongly supports that *C. decandra* is evolutionarily closer to *C. zippeliana* and *A. lanata* is closer to *A. marina*. In addition, two primer pairs were developed for species identification unique to the three *Ceriops* species.

**Abstract:**

*Ceriops* and *Avicennia* are true mangroves in the middle and seaward zones of mangrove forests, respectively. The chloroplast genomes of *Ceriops decandra*, *Ceriops zippeliana*, and *Ceriops tagal* were assembled into lengths of 166,650, 166,083 and 164,432 bp, respectively, whereas *Avicennia lanata* was 148,264 bp in length. The gene content and gene order are highly conserved among these species. The chloroplast genome contains 125 genes in *A. lanata* and 129 genes in *Ceriops* species. Three duplicate genes (*rpl2*, *rpl23*, and *trnM-CAU*) were found in the IR regions of the three *Ceriops* species, resulting in expansion of the IR regions. The *rpl32* gene was lost in *C. zippeliana*, whereas the *infA* gene was present in *A. lanata*. Short repeats (<40 bp) and a lower number of SSRs were found in *A. lanata* but not in *Ceriops* species. The phylogenetic analysis supports that all *Ceriops* species are clustered in Rhizophoraceae and *A. lanata* is in Acanthaceae. In a search for genes under selective pressures of coastal environments, the *rps7* gene was under positive selection compared with non-mangrove species. Finally, two specific primer sets were developed for species identification of the three *Ceriops* species. Thus, this finding provides insightful genetic information for evolutionary relationships and molecular markers in *Ceriops* and *Avicennia* species.

## 1. Introduction

Mangroves are extremely important plants to coastal ecosystems. They protect shorelines from erosion and provide marine nursery areas and breeding sites for a variety of marine and terrestrial organisms (e.g., fish, crustaceans, reptiles, birds, and mammals) [1,2]. For human beings, they are used for food, fuels, timber, and medicines [1,2]. Mangroves grow in the intertidal zones of tropical and subtropical regions with extreme environmental conditions such as frequent tidal inundation, oxygen-poor soil, and high salinity [1,3]. There are roughly 70 mangrove species in 28 genera in 16–19 families [4,5]. Indeed, a few mangrove species in the families Rhizophoraceae and Acanthaceae occupy most areas of mangrove forests [6,7]. In the past decades, mangrove forest areas have been dramatically decreasing due to anthropogenic impacts and climate change [8,9,10]. Therefore, the genetic information of mangroves is crucial for understanding their genetic conservation, population structure, evolution history, and species identification [11,12,13,14]. Recently, a number of whole mangrove genomes have been reported [15,16,17,18,19,20,21,22,23].

*Ceriops* (Rhizophoraceae, Rosids) and *Avicennia* (Acanthaceae, Asterids) are classified as true mangroves and the most dominant species in the middle and seaward zones of mangrove forests, respectively [1,3,24]. Both species have adapted to extreme conditions in mangrove habitats. For example, *Ceriops* is a viviparous mangrove species that has seeds producing propagules or beginning to germinate on the mother plants and is a salt excluder by filtering salt out at the roots [25]. *Avicennia* specially adapted with pneumatophores (pencil-like aerial roots) and salt glands on the upper and lower leaf surfaces, which secrete excess salt from the leaves [26,27]. The genus *Ceriops* contains five species, including *Ceriops australis*, *Ceriops decandra*, *Ceriops pseudodecandra*, *Ceriops tagal*, and *Ceriops zippeliana* [28,29,30,31]. In the past, *C. zippeliana* was believed to be a synonym of *C. decandra* [7,29]; however, differences in morphology and a *trnL* intron of chloroplast DNA between them were reported and they were suggested as different species [31]. *C. decandra* and *C. tagal* are widespread species in a large geographical range from Eastern Africa and throughout tropical Asia and Northern Australia to Melanesia, Micronesia, and Southern China [7,28,29], while *C. pseudodecandra* and *C. australis* are endemic to Australia [32]. *C. zippeliana* is found in Southeast Asia (Thailand, Malaysia, Singapore, Indonesia, and the Philippines) [31]. Based on the International Union for Conservation of Nature (IUCN) Red List, *C. decandra* is classified as a near threatened species due to habitat loss [33]. In addition to *Ceriops*, *Avicennia*, which is the pioneer of the mangrove swamp, comprises at least eight species [7,26,34]. The distribution of *Avicennia* species is separated into two geographic parts: the Indo-West Pacific (IWP) and Atlantic-East Pacific (AEP) regions. At least six species (*A. alba*, *A. integra*, *A. lanata*, *A. marina*, *A. officinalis*, and *A. rumphiana*) are distributed in the IWP region, whereas three species (*A. germinans*, *A. schaueriana*, and *A. bicolor*) are found in the AEP region [26,35,36]. Notably, *A. lanata* is distributed only in Southeast Asia [24]. Recently, *A. bicolor*, *A. integra*, *A. lanata*, and *A. rumphiana* have been listed as vulnerable species on the IUCN Red List [37].

Chloroplasts are photosynthetic organelles in algae and land plants that have their own genomes. Chloroplast genomes are highly conserved because of uni-parent inheritance or maternal inheritance [38]. The sizes of chloroplast genomes in mangrove species are around 145–168 kb [39,40,41,42]. Chloroplast genomes in mangrove species usually contain four regions, including one large single-copy region (LSC), two inverted repeats (IRA and IRB), and one single small-copy region (SSC) [39,40,41,42]. To date, several chloroplast genomes have been reported because of the development of DNA sequencing technology and bioinformatics methods [43,44,45]. For *Ceriops* and *Avicennia* species, only the chloroplast genomes of *C. tagal* and *A. marina* are available [39,41]. Therefore, both the *Ceriops* and *Avicennia* genera suffer from a lack of chloroplast genomes to compare their genomes and to understand phylogenetic relationships among them.

Mangroves present very special ecological characteristics, and understanding the genome structure through this molecular finding will further provide valuable genetic information regarding the evolutionary trends in plants according to harsh climatic conditions. In this study, we investigated the chloroplast genomes of four mangrove species that are commonly distributed in the middle (*Ceriops decandra*, *C. zippeliana*, and *C. tagal*) and seaward (*Avicennia lanata*) zones of the coastal region of Southeast Asia to understand the evolutionary relationships under different coastal environments and to identify genetic markers for species identification and candidate genes under selective pressures. The four mangrove species were sequenced, assembled, and annotated. Comparisons of chloroplast genomes among the three *Ceriops* species and between the *Ceriops* and *Avicennia* species were performed to reveal their evolutionary relationships. Different numbers of SSRs and short repeats were identified among them. Genes under positive selection were identified and might correlate with adaptive selection that could be used for further studies on the response to stress conditions in mangroves. Finally, two sets of species-specific primers were developed for species identification of the three *Ceriops* species based on SSRs. These chloroplast genomes provide valuable genetic information and potential molecular markers for mangrove species in the southeast coastal regions.

## 2. Materials and Methods

### 2.1. Samples, DNA Isolation, and Sequencing

Four mangrove species (*Ceriops decandra*, *Ceriops zippeliana*, *Ceriops tagal*, and *Avicennia lanata*) were used in this study. *C. decandra* is a near threatened species, *C. zippeliana* was formerly recognized as *C. decandra*, and *C. tagal* is a widespread species [7,29]. *A. lanata* is listed as a vulnerable species [37].

Fresh leaves of *C. decandra*, *C. zippeliana*, *C. tagal*, and *A. lanata* were collected from the Ranong, Chanthaburi, Ranong, and Prachuap Khiri Khan provinces in Thailand, respectively (Appendix A). The leaf samples were frozen in liquid nitrogen for DNA isolation. Genomic DNA was extracted using the standard cetyltrimethylammonium bromide (CTAB) method [46]. Each sample was sequenced using the Illumina HiSeqX ten platform with paired-end reads of 150 bp.

### 2.2. Chloroplast Genome Assembly and Annotation

The chloroplast genome of *C. decandra* was assembled using NOVOPlasty version 4.2 [47]. The chloroplast *rbcL* sequence of *C. tagal* (NCBI accession number: MH240830) was used as a seed sequence. The chloroplast genomes of *C. tagal*, *C. zippeliana*, and *A. lanata* were assembled using GetOrganelle [48] with the reference genome-based strategy based on the *C. tagal* chloroplast genome (MH240830) for *Ceriops* species and the *A. marina* chloroplast genome (MT012822) for *Avicennia* species. Notably, GetOrganelle was used mainly for assembling the four mangrove chloroplast genomes due to the highly accurate results of organelle genomes [48,49]. However, it was not fit to complete the chloroplast genome of *C. decandra*; thus, NOVOPlasty was used instead.

All four chloroplast genome sequences were annotated using GeSeq with default settings [50]. The start–stop loci and intron–exon borders of coding genes were edited manually after comparation with reported mangrove chloroplast genes. All transfer RNAs (tRNAs) were predicted using ARAGORN v1.2.36 [51] implemented in the GeSeq software. The circular structures of the chloroplast genomes were illustrated using OGDRAW v1.3. [52]. Finally, the sequences and annotated genes of the four chloroplast genomes were deposited in GenBank (NCBI accession numbers OK258321 (*A. lanata*), OK258322 (*C. tagal*), OK272497 (*C. decandra*), and OK272496 (*C. zippeliana*)).

### 2.3. Comparative Genome Analysis

Comparative genome analysis for *Ceriops* and *Avicennia* was carried out using mVISTA with the Shuffle-LAGAN mode [53]. The species in this analysis included three *Ceriops* species (in this study), *A. lanata* (in this study), and four previously reported mangrove species, including *Kandelia obovata* (NC_042718), *Rhizophora stylosa* (NC_042819), and *Bruguiera parviflora* (MW836113) in the family Rhizophoraceae and *A. marina* (MT012822) in the family Acanthaceae. The previously reported chloroplast genome of *C. tagal* (MH240830.1, China) was used as a reference for comparison [54]. In addition, the junctions and borders of the IR regions were illustrated using IRscope [55].

### 2.4. Repeat and SSR Identification

REPuter [56] was used to identify repeat sequences in the four chloroplast genomes. Furthermore, simple sequence repeats (SSRs) in the chloroplast genome sequences of the three *Ceriops* species and *A. lanata* in this study as well as the previously reported chloroplast genome sequences of *C. tagal* (NCBI: MH240380; CNSA: CNS0105415) and *A. marina* (MT012822 and CNS0105414) were identified using MISA [57]. The thresholds for mononucleotide, dinucleotide, trinucleotide, tetranucleotide, pentanucleotide, and hexanucleotide SSRs were set to 10, 5, 4, 3, 3, and 3, respectively [41]. The minimum distance of compound SSRs was ≤100 bp (default).

### 2.5. Phylogenetic Analysis

To assess the phylogenetic relationships of *Ceriops* and *Avicennia* species, phylogenetic analyses were performed using the maximum likelihood (ML) method based on 50 conserved chloroplast protein-coding genes in 59 plant species, including the 4 species in this study, 19 other mangrove species, 35 relative land plant species, and 1 outgroup species as *Ranunculus macranthus* (NC_008796) (Appendix A). The 50 conserved genes are *atpA*, *atpB*, *atpE*, *atpF*, *atpI*, *ccsA*, *matK*, *ndhA*, *ndhD*, *ndhE*, *ndhG*, *ndhH*, *ndhI*, *ndhK*, *petA*, *petD*, *petG*, *petL*, *petN*, *psaA*, *psaB*, *psaC*, *psaJ*, *psbA*, *psbC*, *psbD*, *psbF*, *psbH*, *psbJ*, *psbL*, *psbM*, *psbN*, *psbT*, *rbcL*, *rpl2*, *rpl14*, *rpl23*, *rpl33*, *rpl36*, *rpoB*, *rpoC1*, *rps2*, *rps3*, *rps4*, *rps8*, *rps11*, *rps12*, *rps14*, *rps15*, and *rps18*. Each gene sequence was aligned individually using MUSCLE with default settings implemented in MEGA X [58]. All gaps in the aligned sequences were removed. The aligned sequences were concatenated in each species. The GTR+I+G model was predicted to be the best fit model for the dataset using the find best DNA/protein model tool in MEGA X. ML analysis was used to construct a phylogenetic tree based on the nucleotide substitution matrix using RAxML version 8.2.10 [59] with the GTRGAMMAI (GTR+I+G) model. Node supports were estimated by performing 1000 bootstrap replicates. Finally, the phylogenetic tree was visualized using FigTree v1.4.3 (http://tree.bio.ed.ac.uk/software/figtree/; accessed on 15 November 2021). Furthermore, gene gain and loss of *rpl32*, *rps16*, and *infA* in mangrove and other non-mangrove species were plotted on the phylogenetic tree.

### 2.6. Gene Selective Pressure Analysis

A total of 61 shared chloroplast protein-coding genes were used to investigate selection pressures for two mangrove genera, *Ceriops* and *Avicennia* (Appendix A). We compared species pairs contained between the three *Ceriops* species and six relative mangrove and non-mangrove species (*Kandelia obovata* (NC_042718), *Rhizophora apiculata* (MW387538), *Bruguiera parviflora* (MW836113), *Pellacalyx yunnanensis* (NC_048998), *Erythroxylum novogranatense* (NC_030601), and *Ranunculus macranthus* (NC_008796)) as well as between *A. lanata* and six relative mangrove and non-mangrove species (*Avicennia marina* (NC_047414), *Coffea arabica* (NC_008535), *Nicotiana tabacum* (NC_001879), *Eucommia ulmoides* (NC_037948), *Lonicera japonica* (NC_026839), and *R. macranthus*). Notably, *R. macranthus* was used as an assumed ancestor for the two mangrove genera. Pairwise sequence alignments for each gene in each species pair were generated using MUSCLE with default settings in MEGA X [58,60]. Then, the values of non-synonymous (Ka) and synonymous (Ks) nucleotide substitutions and Ka/Ks (substitution ratio) in all aligned genes were calculated using KaKs Calculator version 2.0 [61]. Notably, the Ka/Ks ratios were not available (NA) and ~50, indicating no substitution and extremely low Ks values that were replaced to be zero [62]. The Ka/Ks ratios were then visualized using R with the heatmap function [63].

### 2.7. Development of Species-Specific Molecular Markers for Ceriops Species

Two primer pairs were designed from the IR region of the three *Ceriops* chloroplast genome sequences using Primer3 [64]. PCR amplifications were carried out in 20 µL volumes containing 1 µg genomic DNA, 2 µL dNTPs (2.5 mM each), 2 µL of Taq PCR buffer, 0.2 µL of Taq DNA polymerase, and 1.0 µL of each primer. The amplification conditions were 94 °C for 2 min; followed by 30 cycles of 94 °C for 20 s (denaturation), 55 °C for 30 s (annealing), and 72 °C for 30 s (extension); and a final extension of 72 °C for 5 min. PCR products and the DNA ladder were analyzed using a 1% agarose gel to reveal PCR product sizes.

## 3. Results

### 3.1. Chloroplast Genome Features

A total of 66.32 million reads (150 bp) were generated for the three *Ceriops* species and *Avicennia lanata* by the Illumina HiseqX ten platform (Appendix A). These data were used to assemble the four chloroplast genomes with over 300× coverage. The sizes of the complete chloroplast genomes of *C. decandra*, *C. zippeliana*, *C. tagal*, and *A. lanata* were 166,650, 166,083, 164,432, and 148,264 bp in length, respectively (Figure 1 and Table 1). All four species exhibit a typical quadripartite structure, which consists of one large single copy (LSC), one small single copy (SSC), and a pair of inverted repeats (IRs). All four regions of *A. lanata* (LSC: 87,995 bp; SSC: 17,949 bp; IRs: 21,160 bp) were shorter than those of the three *Ceriops* species (92,660–95,217 bp; 18,054–19,158 bp; 26,307–26,535 bp). The overall GC content in the whole chloroplast genomes of the three *Ceriops* species and *A. lanata* was 35% and 38%, respectively. The GC content in the IR regions (~42–44%) was greater than that in the LSC (~32–37%) and SSC (~29–33%) regions.

A total of 125 (*A. lanata*)–129 (*Ceriops* species) genes, including 81–84 protein-coding genes, 36–38 transfer RNA (tRNA) genes, and 8 ribosomal RNA (rRNA) genes, were identified (Table 1 and Table 2). Among them, 14 and 17 genes were duplicated in the IR regions of *A. lanata* and the three *Ceriops* species, respectively (Figure 1 and Table 1 and Table 2). The seventeen genes in the IR regions of the *Ceriops* species were *ndhB*, *rpl2*, *rpl23*, *rps7*, *rps12*, *ycf2*, *rrn4.5*, *rrn5*, *rrn16*, *rrn23*, *trnA-UGC*, *trnE-UUC*, *trnL-CAA*, *trnM-CAU*, *trnN-GUU*, *trnR-ACG*, and *trnV-GAC*. In *A. lanata*, *rpl2*, *rpl23*, and *trnM-CAU* were not found in the IR regions; however, they were located in the LSC region. All photosynthesis genes (45 genes), small subunits of ribosomal proteins (13 genes), DNA-dependent RNA polymerase (4 genes), ribosomal RNA genes (8 genes), other metabolic genes (*matK*, *accD*, *cemA*, *clpP*, and *ccsA*), and conserved open reading frames (*ycf1*, *ycf2*, *ycf3*, and *ycf4*) were found in all four chloroplast genomes. Notably, the *rpl32* gene was lost in *C. zippeliana* and the *infA* gene was only present in *A. lanata*. In *C. zippeliana*, a novel tRNA gene, *trnY-AUA*, was predicted to be located between *trnS-GCU* and *trnT-CGU* of the LSC region. Three genes, *rps16*, *rpl16*, and *ycf2*, were found to be pseudogenes in *A. lanata*, whereas the *rps19* gene was a pseudogene in the *Ceriops* species. Among all the annotated genes, eight protein-coding genes (*atpF*, *ndhA*, *ndhB*, *petB*, *petD*, *rpl2*, *rpl16*, and *rpoC1*) and six tRNA genes (*trnA-UGC*, *trnC-ACA*, *trnE-UUC*, *trnK-UUU*, *trnL-UAA*, and *trnT-CGU*) contain a single intron, and two genes (*rps12* and *ycf3*) contain two introns (Table 2). The largest intron in all species was found in the *trnK-UUU* gene (2504–2585 bp), which contains the *matK* gene.

### 3.2. Comparative Analysis of Chloroplast Genomes

The comparison of eight mangrove chloroplast genomes (three *Ceriops* species, three relative mangrove species in the family Rhizophoraceae, and two *Avicennia* species) showed similar gene organization and variation regions (Figure 2). Gene orientation was assessed among the mangrove species, revealing a conserved gene structure in the chloroplast genomes. Coding regions were more conserved than the non-coding regions. Additionally, IR regions were more conserved than the LSC and SSC regions, suggesting low divergence in the IR regions. The IR regions were highly conserved between *C. tagal* and the Rhizophoraceae species and between *C. tagal* and the *Avicennia* species at over 98% and 90%, respectively. Low similarity (<80%) of nine protein-coding gene sequences (*trnK-UUU*, *trnT-CGU*, *trnL-AAA*, *trnC-ACA*, *rps3*, *rpl22*, *ycf1*, *rps15*, and *rpl32*) was observed between *C. tagal* and the *Avicennia* species. The highly divergent regions were also found in most intergenic regions, especially between the mangrove species in the family Rhizophoraceae and *Avicennia* species.

### 3.3. Chloroplast Boundary Structures

The chloroplast boundary structures of the LSC, SSC, and IRs were compared among the three *Ceriops* species and two *Avicennia* species (Figure 3). In all species, the *ycf1* and *ndhF* genes are located at the boundary of SSC/IRb and SSC/IRa, respectively. The size of *ycf1* is approximately 5800 bp for the *Ceriops* species and 5500 bp for the *Avicennia* species. The ycf1 gene is ~1400 bp away from the SSC/IRb border in the *Ceriops* species, whereas it is ~800 bp away in the *Avicennia* species. Additionally, the size of the *ndhF* gene is similar in all species (2231 bp in *C. tagal*, *A. lanata*, and *A. marina*; 2237 bp in *C. decandra*; and 2243 bp in *C. zippeliana*). The LSC/IRb and LSC/IRa junctions in the three *Ceriops* species positioned the *rps19* gene and *rps19* pseudogene, respectively (Figure 3A). In contrast, the LSC/IRb junction in the two *Avicennia* species positioned the *ycf2* gene (Figure 3B). Notably, no gene stretches across the boundary between the LSC and IRa regions of the two *Avicennia* species. These results reveal that the contraction and expansion of both LSC/IRa and LSC/IRb boundary regions occurred in *Avicennia* and *Ceriops* species, respectively, during their evolution.

### 3.4. Chloroplast Repeats and SSRs

Repeats in the chloroplast genomes of the three *Ceriops* species and *A. lanata* were identified (Figure 4A–C and Appendix A). The number of forward, reverse, palindromic, and complement repeats was different in each species. For example, 23, 35, 25, and 17 forward repeats were found in *C. decandra*, *C. zippeliana*, *C. tagal*, and *A. lanata*, respectively (Figure 4A). The number of forward repeats in *C. zippeliana* (35) was the highest, while the number of palindromic repeats in *A. lanata* (20) was the highest (Figure 4A). There was no complement repeat in *C. zippeliana*. Interestingly, all *Ceriops* species contained long repeats (>30 bp), whereas *A. lanata* species consisted of short repeats (<40 bp) (Figure 4B). Usually, most repeats in all species were observed in the LSC region (Figure 4C).

SSRs in the chloroplast genomes of *Ceriops* species, *A. lanata*, and related mangrove species were analyzed (Figure 4D–F and Table 3 and Appendix A). Mononucleotide SSRs were the most prevalent in all species (Figure 4D), consisting predominantly of A/T repeats, at over 90% (Figure 4E). Most SSRs were found in the LSC region (Figure 4F). Some SSRs were unique in each *Ceriops* species.

### 3.5. Ceriops Species Identification Based on Species-Specific Molecular Markers

Two pairs of primers were designed and tested to identify the differences between *Ceriops* species. PCR products of the chloroplast genomes exhibited different sizes among the three *Ceriops* species based on one molecular marker using two primer pairs (Figure 5). The PCR product of the two primer pairs confirmed the same variation. For the first one, the PCR product sizes of *C. tagal*, *C. decandra*, and *C. zippeliana* were 167, 207, and 226 bp, respectively (Figure 5 and Appendix A). For the other one, the PCR product sizes of *C. tagal*, *C. decandra*, and *C. zippeliana* were 323, 363, and 382 bp, respectively (Figure 5 and Appendix A). The difference in PCR product sizes occurred from indels and SSRs in the IR regions. For example, the chloroplast sequence of these regions of *C. tagal* and *C. zippeliana* contained 18 and 5 dinucleotide (AT/TA) repeats, respectively (102,313–102,348 and 154,744–154,779 bp: *C. tagal*; 120,115–120,124 and 156,400–156,409 bp: *C. zippeliana*) (Appendix A), whereas there were no SSRs in these regions of *C. decandra* based on the SSR analysis criteria in this study due to short (AT/TA) repeats (<4 repeats) (Appendix A).

### 3.6. Phylogenetic Relationships

The maximum likelihood (ML) analysis, based on 50 conserved chloroplast genes in 59 plant species, resulted in the best single tree (Figure 6). The ML tree shows two major clades corresponding to Rosids and Asterids. This tree highly supports that all *Ceriops* species are in the family Rhizophoraceae (Rosids), whereas *A. lanata* is in the family Acanthaceae (Asterids). *C. decandra* is closely related to *C. zippeliana* with a monophyletic branch supported by 100% bootstrap values. *C. tagal* is a sister species of the other two *Ceriops* species. For other mangrove species in the family Rhizophoraceae, *Kandelia obovata* is closer to the *Ceriops* species than *Rhizophora* and *Bruguiera* species. In addition, *A. lanata* and *A. marina* are grouped together in the family Acanthaceae (Asterids), with a bootstrap value of 100%.

The gain and loss of the *rpl32*, *rps16*, and *infA* genes in mangrove and non-mangrove species were plotted in the phylogenetic tree (Figure 6). For example, the *rpl32* gene was lost in four mangrove species in Rhizophoraceae, namely *C. zippeliana*, *K. obovata*, *Rhizophora stylosa*, and *Bruguiera gymnorhiza*. The *rps6* gene was lost in all mangrove species in Rhizophoraceae and most land plant species in Malpighiales, but not in Acanthaceae (Lamiales). The *infA* gene was also lost in all mangrove species in Rhizophoraceae and most land plant species in Rosids.

### 3.7. Chloroplast Genes under Positive Selection

To identify candidate genes under positive selection, the values of Ka/Ks (non-synonymous/synonymous) were estimated for 61 conserved chloroplast protein-coding genes in *Ceriops* and *Avicennia* species to relative mangrove species and non-mangrove species (assumed ancestors) (Figure 7 and Appendix A). Most Ka/Ks ratios were lower than 1.0. However, there were two genes, *rps7* and *rps15*, in which the Ka/Ks ratios were greater than 1.0 in several compared species pairs, suggesting positive selection during their evolution. The *rps7* gene was under positive selection in both *Ceriops* species and *A. lanata* compared with relative non-mangrove species. The average Ka/Ks ratio of the *rps7* gene between the *Ceriops* species compared with *Ctenolophon englerianus*, *Averrhoa carambola*, and *Vitis rolundifolia* was 1.06, 1.11, and 1.81, respectively. The Ka/Ks ratio of the *rps7* gene between *A. lanata* compared with *Eucommia ulmoides* and *Lonicera japonica* was 1.03 and 1.13, respectively. In addition, the *rps15* gene was positively selected in *C. decandra* and *C. zippeliana*. The Ka/Ks average ratio of the *rps15* gene between the two *Ceriops* species compared with *Pellacalyx yunnanensis*, *B. parviflora*, and *R. apiculata* was 1.16, 1.17, and 1.42, respectively.

## 4. Discussion

Diverse chloroplast genome sequences have been used to study the evolution of mangrove species and to identify different mangrove species [39,40,41,42]. In the current study, we reported the chloroplast genomes of four mangrove species, including three *Ceriops* species (*C. decandra*, *C. zippeliana*, and *C. tagal*) and *Avicennia lanata.* Based on morphological characteristics, *Ceriops* is classified to the family Rhizophoraceae of the order Rosids (polypetalous), whereas *Avicennia* belongs to the family Acanthaceae of the order Asterids (sympetalous) [1,3]. *Ceriops* and *Avicennia* have a convergent evolution and are the most dominant species in the middle and seaward zones of mangrove forests, respectively [1,3,24]. The three *Ceriops* chloroplast genomes (164.4–166.7 kb) were slightly different, consistent with published chloroplast genomes of mangrove species (middle zone) in Rhizophoraceae such as *C. tagal*, *Kandelia obovata*, *Rhizophora* species, and *Bruguiera* species (160.3–164.6 kb) [40,41,42,65,66,67]. In contrast, the smaller chloroplast genome of *A. lanata* was 148.2 kb, which is similar to the previously reported chloroplast genome of *Avicennia marina* (147.9–152.3 kb) [39,68]. In addition, the chloroplast genomes of *Sonneratia alba* and *Sonneratia apetala*, which are true mangroves in the family Lythraceae of the order Rosids in the seaward zone, were approximately 153.1 kb [69,70]. This finding suggests that the size of mangrove chloroplast genomes in the seaward zone may be compact compared with mangrove species in the middle zone, which is caused by adaptation under coastal stress conditions, especially salt stress. Salinity can affect plants in several ways, such as by changing the chloroplast size, number, lamellar organization, and lipid and starch accumulation and interfering with cross-membrane transportation [71].

Chloroplast genomes are usually conserved in genome organization, gene order, and gene content [72]. Nevertheless, gene gain and loss have been found among the four mangrove species. The *infA* gene (translation initiation factor 1) was found in *A. lanata* but not in the *Ceriops* species and other mangrove species in Rhizophoraceae [40,41,42]. The loss of the *infA* gene from the chloroplast to the nucleus occurred independently in multiple angiosperm lineages, especially in Rosids [73,74]. The *rpl16* and *rps16* genes became pseudogenes in *A. lanata* but not in *A. marina* [39]. The *rpl16* gene has been independently pseudogenized in several angiosperm lineages across eudicots and monocots [75,76,77]. Notably, the *rps16* gene was not found in the three *Ceriops* species, consistent with other mangrove and land plant species in the order Malpighiales [40,42,78]. The *rps16* gene has been a pseudogene or lost by the nuclear encoded *rps16* in many higher plants [79,80,81,82]. Three genes, namely *rpl2*, *rpl23*, and *trnM-CAU*, retained one copy in the LSC region of *A. lanata* and were found in only a single copy in the LSC region of *A. marina* [39]. In contrast, the three genes are located in the IR regions in the *Ceriops* species; thus, they have two copies, concordant with other mangrove species in Rhizophoraceae [40,41,42]. Contraction at the LSC/IR junction, which was observed in several land plants, might result in the deletion of *rpl2* and *rpl23* from one of the IR regions [83,84]. Remarkably, *rpl32* was lost in *C. zippeliana* but not the other *Ceriops* species and *A. lanata*. The loss of *rpl32* has occurred in many mangrove species in the family Rhizophoraceae, such as *K. obovata*, *R. stylosa*, and *B. gymnorhiza* [40,42]. Transfer of chloroplast *rpl32* to the nucleus DNA occurred independently in several families of Malpighiales plants, such as Rhizophoraceae, Erythroxylaceae, Ctenolophonaceae, Violaceae, Passifloraceae, Salicaceae, and Euphorbiaceae [40,42,78,85,86,87,88]. These reveal gene evolution in *Ceriops*, *Avicennia*, and other mangrove species.

The border positions of the LSC, SSC, and IR regions were compared among the *Ceriops* and *Avicennia* chloroplast genomes. The boundaries of the LSC/IRa and LSC/IRb regions between the *Ceriops* species and *A. lanata* had different gene positioning. In the *Ceriops* species, the *rps9* gene was located at the LSC/IRb border and the *rps9* pseudogene was located at the LSC/IRa border, concordant with other mangrove species such as *Bruguiera* species [42]. Meanwhile, the *ycf2* gene was located at the LSC/IRb border in *A. lanata* and no gene was located at the LSC/IRa border, which was similar with *A. marina* and some non-mangrove species in the Acanthaceae, such as *Ruellia breedlovei* (KP300014) [89,90]. One of the reasons for chloroplast genome variation among angiosperms is the contraction or expansion of the IR regions [91]. These indicated that the contraction of the IR regions in *A. lanata* and the expansion of the IR regions in the *Ceriops* species may be mainly caused by decreasing and increasing gene duplications in the IR regions, respectively, during their evolution.

Repeats of the four mangrove species varied among them. The occurrence of short repeats (<40 bp) and a small number of SSRs was found in *A. lanata* due to a compact chloroplast genome containing small non-coding regions. The mangrove species carry mostly forward repeats in their chloroplast genomes that are similar in other mangrove chloroplast genomes [39,41,42]. For SSRs, most consist of repetitions of an A/T mononucleotide in all four mangrove species, concordant with other mangrove species [39,41,42]. SSRs with repeat length differences occur from the process of mutation [92], which could be used for identifying related species. In general, differentiation between *C. decandra* and *C. zippeliana* based on morphology can be difficult; therefore, we designed two species-specific primer sets based on one different SSR in the IR regions among the three *Ceriops* species. The specific primer sets were tested and could be used to identify *C. decandra* and *C. zippeliana* as well as *C. tagal*.

*C. decandra* is more closely related to *C. zippeliana* than to *C. tagal*, concordant with the results based on morphological and molecular evidence as well as the phylogenetic tree based on the *trnL* intron sequence of chloroplast genomes [30]. The *Ceriops* species are more closely related to *K. obovata* than other mangrove species in Rhizophoraceae, consistent with the results based on 44 conserved genes in 71 species (14 mangrove species and 57 land plant species) using Bayesian inference (BI) and ML [41]. In addition, the chloroplast genome of two *Avicennia* species was shown to be closely related to Acanthaceae, with a bootstrap value of 100%. These two lineages (*Ceriops* and *Avicennia*) had paraphyletic clades of the phylogeny, indicating convergent evolution.

The low average Ka/Ks ratios of most conserved genes in the four mangrove species suggest that the whole-chloroplast protein level of the species has been subjected to strong purifying selections. In general, synonymous changes (Ks) occur more often than non-synonymous substitutions (Ka); as a result, the ratios of Ka/Ks are commonly lower than 1.0 [93]. Remarkably, the Ka/Ks ratios of two chloroplast genes (*rps7* and *rps15*) were greater than 1.0, suggesting positive selection pressure. The *rps7* gene encodes ribosomal protein S7 involved in the regulation of chloroplast translation [94]. Positive selection on the *rps7* gene has also been observed in many mangrove species (such as *K. obovata*, *Rhizophora* species, and *Bruguiera* species) and some land plants (such as *Ananas comosus* (pineapple)) [42,95,96]. Moreover, the *rps15* gene encoding ribosomal protein S15 was under positive selection in *C. decandra* and *C. zippeliana*. The multiple sequence alignment result showed co-variation of three sites in the *rps15* amino acid sequence that occurs in *C. decandra* and *C. zippeliana* but not in *C. tagal* (Appendix A). Interestingly, one amino acid site at position 75 was unique in only *C. decandra* and *C. zippeliana* (Isoleucine) compared with mangrove and non-mangrove species (Valine) in the family Rhizophoraceae. The *rps15* gene was also reported to be related to evolution under positive selection in Araliaceae species [97]. Knockout of the chloroplast *rps15* gene in tobacco leads to a specific reduction in small 30S ribosomal subunits [98]. Thus, these genes, *rps7* and *rps15*, might be undergoing adaptive evolution in response to stress environments in mangrove forests.

## 5. Conclusions

In this study, the complete chloroplast genome sequences of *Ceriops decandra*, *C. zippeliana*, *C. tagal*, and *Avicennia lanata* were sequenced and compared. The chloroplast genome of *A. lanata* (seaward zone) is compact compared with the three *Ceriops* species (middle zone). The chloroplast genomes are mostly conserved in genome organization, gene order, and gene content; however, gene gain and loss have been found among them. The occurrence of contraction or expansion of IR regions in *Avicennia* and *Ceriops* species would be a result of decreasing and increasing gene duplications in the IR regions, respectively. Phylogenetic analysis showed that *C. decandra* is closer to *C. zippeliana* than to *C. tagal* in the family Rhizophoraceae, and *A. lanata* is clustered with *A. marina* in the family Acanthaceae, which supports convergent evolution between the two genera. The different chloroplast repeats and SSRs in the four mangrove species can be used as genetic markers, and two species-specific primer sets have been developed for species identification among the three *Ceriops* species in this work. The *rps7* gene was identified under positive selection among mangrove species and might correlate with adaptive selection under coastal environments. Hence, these results could not only provide valuable genetic information of mangrove *Ceriops* and *Avicennia* species but also offer molecular markers for species identification and a candidate gene in response to climatic stress conditions of coastal environments.

## Figures and Tables

**Figure 1 biology-11-00383-f001:**
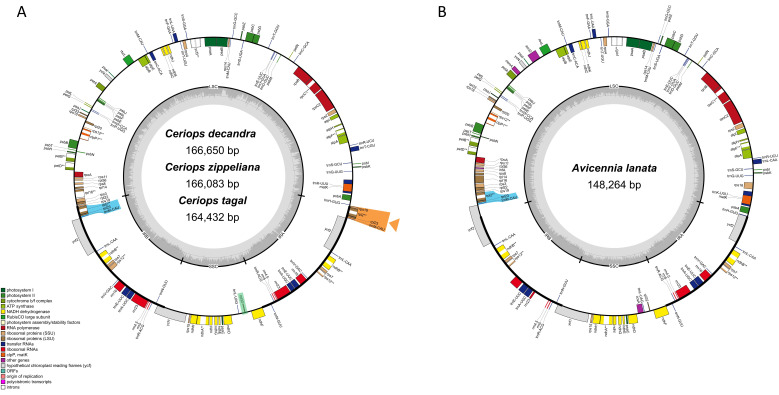
The chloroplast features of four mangrove species. (**A**) Complete chloroplast maps of *Ceriops decandra*, *Ceriops zippeliana*, and *Ceriops tagal*. (**B**) Complete chloroplast map of *Avicennia lanata*. Genes located outside and inside the circle are transcribed clockwise and counter-clockwise, respectively. The grey bar area in the inner circle indicates GC content of the genome, whereas the lighter grey area indicates AT content of the genome. LSC, SSC, and IRs (IRA and IRB) represent large single copy, small single copy, and inverted repeats, respectively. Genes based on different functional groups are shown in different colors. Green rectangle indicates a loss region (*rpl32*) of *C. zippeliana*. Labeling in blue color indicates the same region of three genes (*rpl2*, *rpl23*, and *trnM-CAU*) in both *Ceriops* and *Avicennia* species, whereas labeling in orange color with an orange arrow indicates a unique region of three duplicate genes (*rpl2*, *rpl23*, and *trnM-CAU*) in three *Ceriops* species compared to *A. lanata*. ** indicates genes containing introns.

**Figure 2 biology-11-00383-f002:**
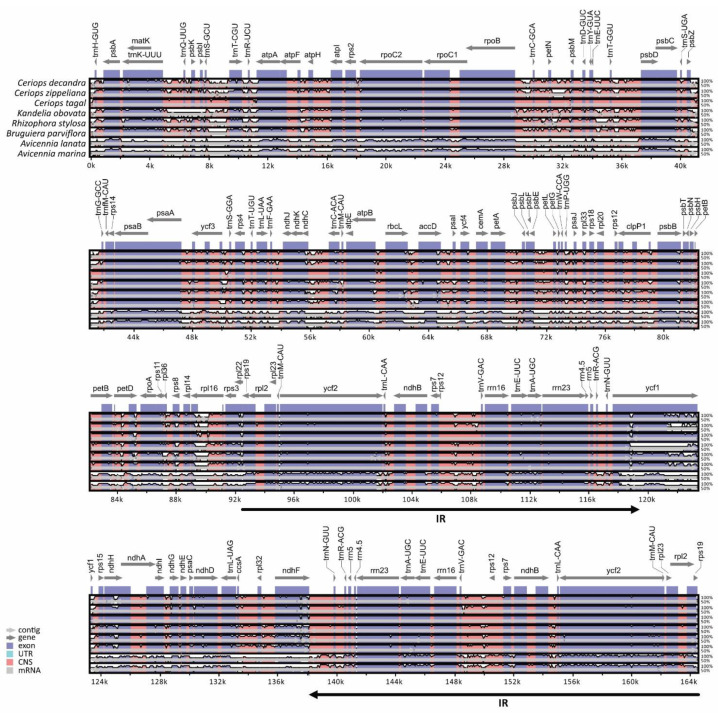
Alignment map of 8 mangrove chloroplast genomes in Rhizophoraceae and Acanthaceae. Horizontal axis is genome position, whereas vertical axis shows sequence identity. Grey arrows indicate genes and transcriptional directions, and black arrows indicate IR regions. Exons and conserved non-coding sequences (CNS) are shown in blue and red, respectively.

**Figure 3 biology-11-00383-f003:**
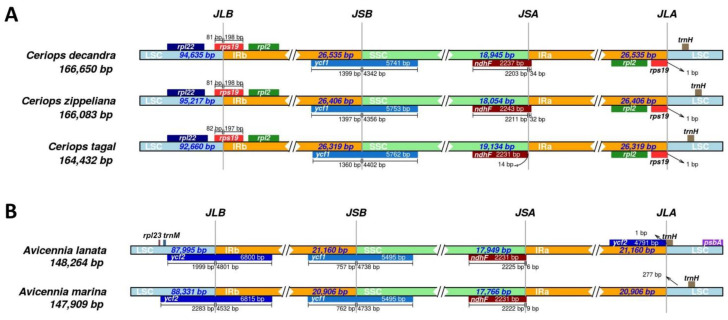
Comparison of IR boundaries of chloroplast genomes. (**A**) IR boundaries among three *Ceriops* species. (**B**) IR boundaries among two *Avicennia* species.

**Figure 4 biology-11-00383-f004:**
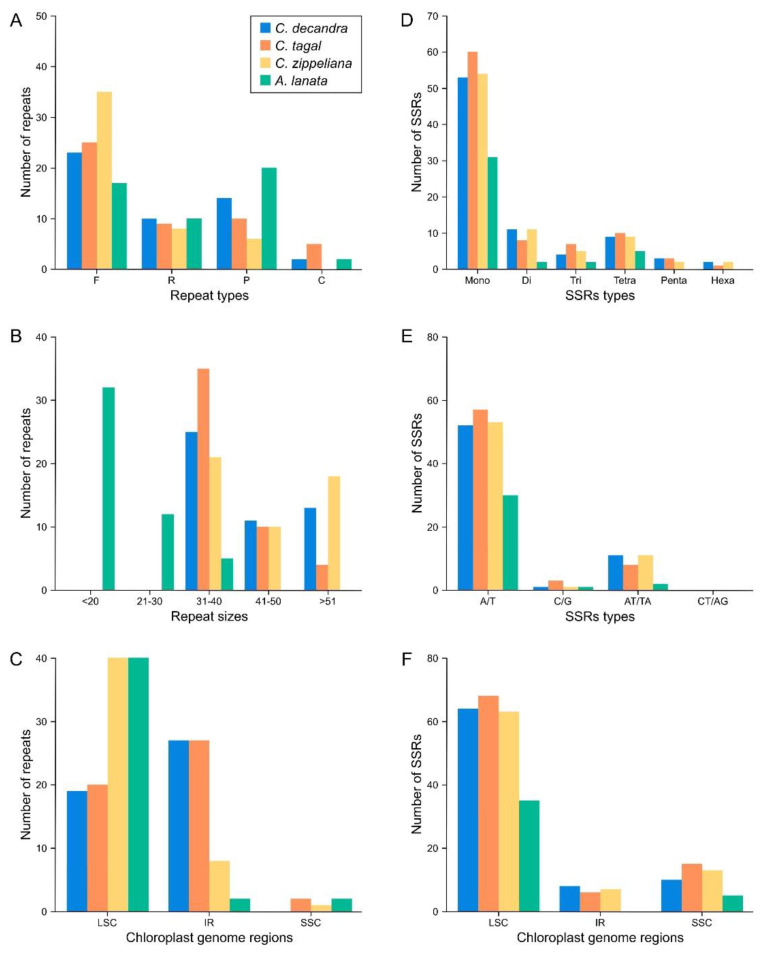
Statistical analysis of repeats and SSRs in four mangrove chloroplast genomes. (**A**) Sorted by type of repeat. (**B**) Frequency by repeat types. (**C**) Sorted by repeat region of genome. (**D**) Sorted by type of SSR. (**E**) Frequency by SSR type. (**F**) Sorted by SSR region of genome.

**Figure 5 biology-11-00383-f005:**
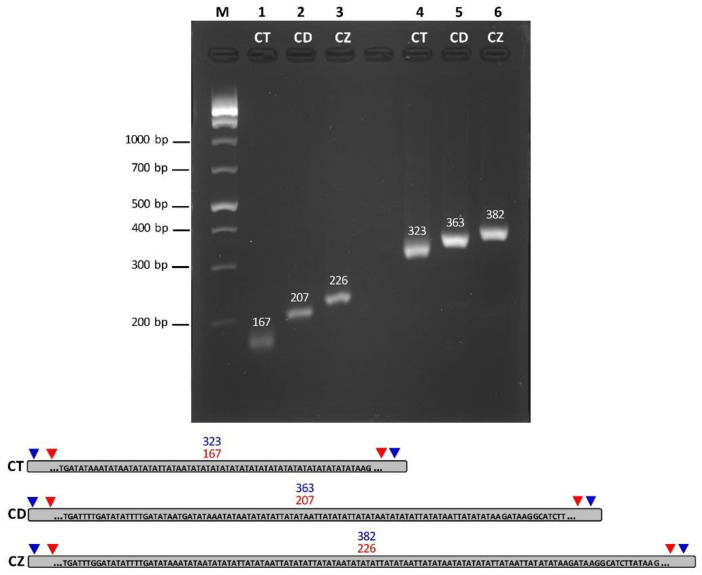
Analysis of PCR products by 1% agarose gel electrophoresis with graphical genomes positing primer pairs. PCR products were amplified with two primer pairs in three *Ceriops* chloroplasts. Lane M: DNA size marker; Lane 1: PCR product amplified with the first primer set in *C. tagal* (CT: 167 bp); Lane 2: PCR product amplified with the first primer set in *C. decandra* (CD: 207 bp); Lane 3: PCR product amplified with the first primer set in *C. zippeliana* (CZ: 226 bp); Lane 4: PCR product amplified with the second primer set in *C. tagal* (323 bp); Lane 5: PCR product amplified with the second primer set in *C. decandra* (363 bp); Lane 6: PCR product amplified with the second primer set in *C. zippeliana* (382 bp). Graphical genomes show a part of the IR region used for designing specific primers of three *Ceriops* species based on different SSRs. Red arrows indicate the position of the first primer set, whereas blue arrows indicate the position of the second primer set.

**Figure 6 biology-11-00383-f006:**
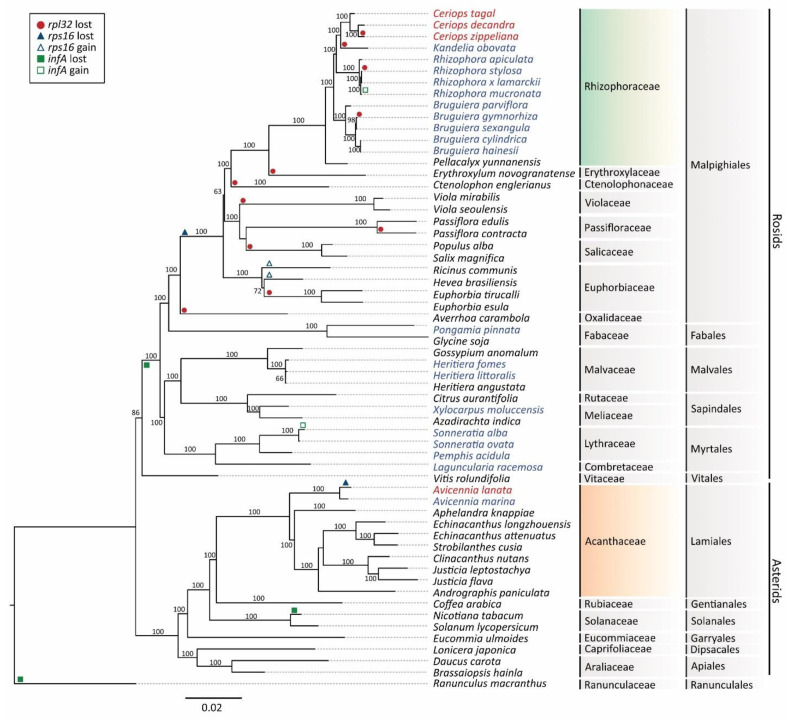
Maximum likelihood (ML) tree for 50 chloroplast protein-coding genes in 59 plant species. Values above the branches represent bootstrap with 1000 replicates. The mangrove species in this study are indicated in red text, whereas other mangrove species are indicated in blue text. The Rhizophoraceae lineage is indicated in gradient green, and the Acanthaceae lineage is indicated in gradient orange. Gain and loss of the *rpl32*, *rps16*, and *infA* genes are shown in different symbols and colors.

**Figure 7 biology-11-00383-f007:**
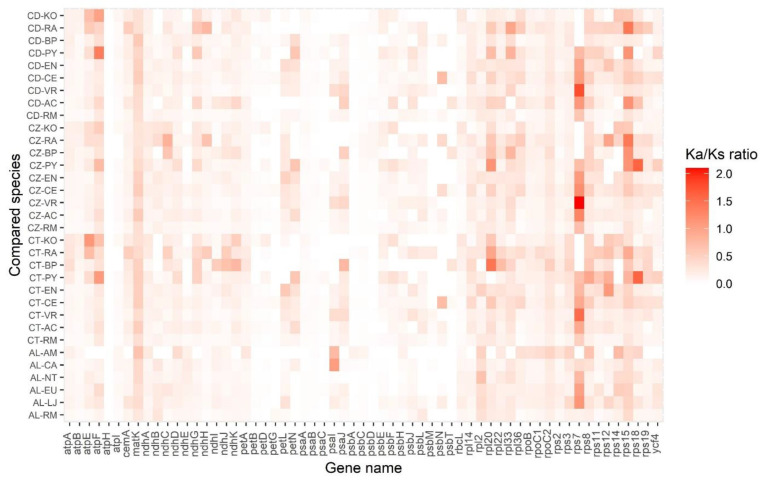
Heatmap of Ka/Ks ratios between every compared species in 61 chloroplast genes. The scale ratios associated with each value are shown in the key beside the figure. AL (*Avicennia lanata*), AM (*Avicennia marina*), CA (*Coffea arabica*), EU (*Eucommia ulmoides*), LJ (*Lonicera japonica*), NT (*Nicotiana tabacum*), RM (*Ranunculus macranthus*), CD (*Ceriops decandra*), CZ (*Ceriops zippeliana*), CT (*Ceriops tagal*), AC (*Averrhoa carambola*), BP (*Bruguiera parviflora)*, CE (*Ctenolophon englerianus)*, EN (*Erythroxylum novogranatense*), KO (*Kandelia obovata*), PY (*Pellacalyx yunnanensis*), and RA (*Rhizophora apiculata*).

**Table 1 biology-11-00383-t001:** Summary of the chloroplast genomes of three *Ceriops* species and *Avicennia lanata*.

Genome and Gene Features	*C. decandra*	*C. zippeliana*	*C. tagal*	*A. lanata*
Genome size (bp)	166,650	166,083	164,432	148,264
LSC (bp)	94,635	95,217	92,660	87,995
SSC (bp)	18,945	18,054	19,158	17,949
IR (bp)	26,535	26,406	26,307	21,160
LSC GC content (%)	32.07	31.67	32.59	36.77
SSC GC content (%)	29.28	29.63	29.31	32.72
IR GC content (%)	41.90	41.86	42.17	44.26
Genome GC content (%)	34.89	34.69	35.28	38.42
No. of total genes	129	129	129	125
No. of protein coding genes	84	83	84	81
No. of rRNAs	8	8	8	8
No. of tRNAs	37	38	37	36
No. of duplicated genes	17	17	17	13
Pseudogenes	1 (*rps19*)	1 (*rps19*)	1 (*rps19*)	3 (*rpl16*, *rps16, ycf2*)
Gene gain/loss	-	−*rpl32*	-	+*infA*

**Table 2 biology-11-00383-t002:** List of annotated genes in the chloroplast genomes of three *Ceriops* species and *Avicennia lanata*.

Category	Group of Genes	Gene Name
Photosynthesis	Subunits of Photosystem I	*psaA*, *B*, *C*, *I*, *J*
	Subunits of Photosystem II	*psbA*, *B*, *C*, *D*, *E*, *F*, *H*, *I*, *J*, *K*, *L*, *M*, *N*, *T*, *Z*
	Subunits of NADH dehydrogenase	*ndhA **, *B **(×2), *C*, *D*, *E*, *F*, *G*, *H*, *I*, *J*, *K*
	Cytochrome b6/f complex	*petA*, *B **, *D **, *G*, *L*, *N*
	ATP synthase	*atpA*, *B*, *E*, *F **, *H*, *I*
	Rubisco	*rbcL*
Self-replication	Large subunit of ribosomal proteins	*rpl2 **(×2), *14*, *16 **^, e^, *20*, *22*, *23*(×2), *32*^b^, *33*, *36*
	Small subunit of ribosomal proteins	*rps2*, *3*, *4*, *7*(×2), *8*, *11*, *12 ***(×2), *14*, *15*, *16*^d^*, 18*, *19*
	DNA dependent RNA polymerase	*rpoA, B*, *C1 **, *C2*
	rRNA genes	*rrn4.5*(×2), *5*(×2), *16*(×2), *23 **(×2)
	tRNA genes	*trnA-UGC **(×2), *trnC-ACA* *, *trnC-GCA*, *trnD-GUC*,
		*trnE-UUC **(×3), *trnF-GAA*, *trnG-GCC, trnH-GUG*,
		*trnK-UUU **, *trnL-CAA*(×2), *trnL-UAA **, *trnL-UAG*,
		*trnM-CAU*(×4), *trnN-GUU*(×2), *trnP-UGG*,
		*trnQ-UUG*, *trnR-ACG*(×2), *trnR-UCU*, *trnS-GCU*,
		*trnS-GGA*, *trnS-UGA*, *trnT-CGU **, *trnT-GGU*,
		*trnT-UGU*, *trnV-GAC*(×2), *trnW-CCA*, *trnY-AUA*^a^
		*trnY-GUA*
Other genes	Maturase	*matK*
	Subunit Acetyl-CoA-Carboxylate	*accD*
	Envelop membrane protein	*cemA*
	Protease	*clpP ***
	C-type cytochrome synthesis gene	*ccsA*
	Translation initiation factor gene	*infA* ^d^
Unknown	Conserved open reading frames	*ycf1*, *2* (×2) ^e^, *3 ***, *4*
	Pseudogene	*rps19*^c^, *rps16*^f^*, rpl16*^f^*, ycf2*^f^

Notes: * Gene with one intron. ** Gene with two introns. ^a^ Gain gene in *C. zippeliana.*
^b^ Loss gene in *C. zippeliana.*
^c^ Pseudogene in *Ceriops* species. ^d^ Gain gene in *A. lanata.*
^e^ Loss gene in *A. lanata.*
^f^ Pseudogene in *A. lanata.*

**Table 3 biology-11-00383-t003:** Number of SSRs in the chloroplast genomes of three *Ceriops* and two *Avicennia* species.

Species	SSR Type						Total Number	The Number of SSRs for Compound Formation
	Mono-	Di-	Tri-	Tetra-	Penta-	Hexa-		
CD	74	25	14	21	5	3	142	36
CZ	78	40	21	29	7	2	177	64
CT	81	17	20	23	4	1	146	35
CT ^a^	79	16	20	22	4	1	142	35
CT ^b^	78	16	20	23	4	1	142	35
AL	38	2	3	7	0	0	50	5
AM ^a^	34	2	2	6	0	0	44	1
AM ^b^	49	1	4	7	0	0	61	4

Notes: CD: *Ceriops decandra* (OK272497); CZ: *C. zippeliana* (OK272496); CT: *C. tagal* (OK258322); CT ^a^: *C. tagal* (MH240380); CT ^b^: *C. tagal* (CNS0105415); AL: *Avicennia lanata* (OK258321); AM ^a^: *A*. *marina* (MT012822); AM ^b^: *A. marina* (CNS0105414).

## Data Availability

The chloroplast genome sequences of *Ceriops decandra*, *Ceriops zippeliana*, *Ceriops tagal*, and *Avicennia lanata* were submitted to the National Center for Biotechnology Information (NCBI), with the accession numbers OK272497, OK272496, OK258322, and OK258321, respectively.

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
