# Peer review of "Comparative Analysis and Phylogenetic Relationships of Ceriops Species (Rhizophoraceae) and Avicennia lanata (Acanthaceae): Insight into the Chloroplast Genome Evolution between Middle and Seaward Zones of Mangrove Forests"

_biology, 2022, doi:10.3390/biology11030383_

Round 1

Reviewer 1 Report

Mangrove presents very special ecological characteristics and understanding the species genome structure and build-up through molecular findings can help inform how plants adapt to harsh climatic conditions. This comparison  among and between mangrove species of the middle and seaward zones will be more informative on habitat and evolutional trends. The authors used appropriate approach and right tools to compare the chloroplast genome for species of Ceriops and Avicennia genera. In view of quality improvement and consideration for publication, I will request the authors to react to the comments and suggestions presented here in.

General comments:

1- The introduction and discussion sections need moderate English editing regarding the style.

2- Can the authors briefly comment on their choice of samples e.g. the genus Avicennia got more than two species...why just one was chosen while three species for Ceriops?

3- NOVOplasty was used for Ceriops while GetOrganelle for Avicennia genome assembly. Why not same program? What are the strength and weaknesses of these programs? Can this influence the out put of the assembly results?

4- Digital object identifier (DOI) e.g reference 27 and ISSN number (25 and 28) should be included in references where applicable

5- I proposed editing figure 1 and 2 for clear visualization.

6- Consider text justification for some of the sections on pages 9,10,12,13 and14.

7- The lost of rpl32 from C. zippeliana, rpl16, rps16 changed to pseudogenes and single copy of rpl2, rpl23 and trnM-CAU retained in A. lanata are some of the major findings upon comparing chloroplast genome. However, in the discussion section there is lack of sustainable information linking this to the evolutional trend. It will be good for such argument to be included.

Specific comments:

1-Line 55:...regions that are... (delete "that are" replace using the word "with")

2-Lnes 57-58: Rephrase...Indeed, a few mangrove species especially in the family ...occupy most area of the mangrove.

3-Line 59: In the past decades,...

4-Lines 61-62: Therefore,... is crucial in understanding genetic conservation, population structure, evolutional history and species identification.

5-Lines 78-79: ...while C pseudodecandra and C. australis are endemic to Australia C. zippeliana is found in...

6-Lines 96: I propose deleting "In the past decade" and use "To date,"

7-Line 171: A total "of " 61...

8-Line 217...of three genes (...) in both Ceriops and Avicennia species, whereas...

9-Line 221: Table 1.

Under the species column is a list of the different genes of chloroplast genome. A better way of presenting this data might be to consider placing the genome and genes features in same row as species. 

10-Line 240: Insert "," between "intron"  and "and"... 

11-Lines 264: ...vertical axis shows sequence identity...

12-Lines 376-382: ...ratios between 61species compared chloroplast genes. In addition, it is some how not easy for the reader to link the initial for species name to corresponding species. I proposed stating the species in bracket following each and corresponding string of initials.

13-Lines 325-331: A photograph of 1% agarose gel showing result of amplified PCR product.

"The authors mentioned gel purify PCR product" ..does it mean the PCR product was gel purified before loading on an agarose gel for electrophoresis?

The legend of this figure can be rephrase in a way to limit the frequency of using "PCR product amplified with"........

In my opinion, for clarity and for better understanding to the reader...adding graphic picture illustrating the regions (genome-sequence) used for primer designing and differentiation of these species side by side the gel picture.

14-Lines 348-349: ...indicated in green shade OR shaded green (choose)

Author Response

Dear Reviewer,

Best regards,

Panthita

Reviewer 2 Report

The manuscript entitled "Comparative analysis and phylogenetic relationships of Ceriops species (Rhizophoraceae) and Avicennia lanata (Acanthaceae): insight into the chloroplast genome evolution between middle and seaward zones of mangrove forests" submitted to my revision presents results of study on plastome variability and evolution in four mangrove species, representing two plant families. The study reported in the manuscript is well planned, thought out and well described. The scientific language used is clear and understandable, the introduction presents the research object well, the methodology is comprehensive, the results are comprehensive and clearly presented. The text is prepared with great care, as evidenced by the fact that it is difficult to find linguistic errors, typos or editorial errors. The presented graphics illustrate the results well, they are understandable and aesthetic. I consider the topic raised in the research to be justified, and the tested object worthy of attention. I highly appreciate the work, although I have two remarks that could further improve the quality of the publication and make it even more transparent and understandable for the reader. 

The first issue is the Intdoduction.  I would reccommend the Authors to consider highlighting the purpose of the research more clearly. I mean mainly justifying the application of the selection pressure analysis. It would be helpful to formulate a research hypothesis here. I would also suggest a clearer justification for the choice of species for the study.

The second comment concerns the first paragraph of the Discussion, where the size of the cp genome is discussed . I would suggest the slight reedition of the text to emphasize the observed relation of genome size with sample location. Otherwise, the reader might be confused as the examples discussed belong to different orders and even families, they also are mangrove or non-mangrove and also two lacations are discussed. This might be a little bit confusing and forces the reader to study the paragraph several times to notice a certain relationship indicated by the Authors further. 

My rate of the manuscript is high and I recommend to accept it after minor revision.

Author Response

Dear Reviewer,

Best regards,

Panthita
